# Cyclical Trends of Network Load Fluctuations in Traffic Jamming

Bosiljka Tadić [1,2] 

1   Department of Theoretical Physics, Jožef Stefan Institute, 1001 Ljubljana, Slovenia; bosiljka.tadic@ijs.si
2   Complexity Science Hub, Josephstaedterstrasse 39, 1080 Vienna, Austria

**Abstract:** The transport of information packets in complex networks is a prototype system for the study of traffic jamming, a nonlinear dynamic phenomenon that arises with increased traffic load and limited network capacity. The underlying mathematical framework helps to reveal how the macroscopic jams build-up from microscopic dynamics, depending on the posting rate, navigation rules, and network structure. We investigate the time series of traffic loads before congestion occurs on two networks with structures that support efficient transport at low traffic or higher traffic density, respectively. Each node has a fixed finite queue length and uses next-nearest-neighbour search to navigate the packets toward their destination nodes and the LIFO queueing rule. We find that when approaching the respective congestion thresholds in these networks, the traffic load fluctuations show a similar temporal pattern; it is described by dominant cyclical trends with multifractal features and the broadening of the singularity spectrum regarding small-scale fluctuations. The long-range correlations captured by the power spectra show a power-law decay with network-dependent exponents. Meanwhile, the short-range correlations dominate at the onset of congestion. These findings reveal inherent characteristics of traffic jams inferred from traffic load time series as warning signs of congestion, complementing statistical indicators such as increased travel time and prolonged queuing in different transportation networks.

**Keywords:** information transport on complex networks; critical dynamics; traffic jamming; multifractal fluctuations; long-range correlations; power spectra

## 1. Introduction: Traffic of Information Packets on Complex Networks

Studies of the diffusion of information packets on the Internet (see [1–3] and references therein) provide a theoretical framework towards understanding traffic jamming. Jamming is a fascinating nonlinear dynamic phenomenon occurring in different systems, from granular flow in materials to large-scale transportation systems such as city vehicles traffic and aerospace transport [4–10]. A line of research consists of macroscopic density dynamic models [4,11,12] originally designed to describe urban vehicles transport and understand the global dynamic properties of jamming. On the other hand, microscopic modelling and simulations of traffic on different network topologies have provided ways to see how local properties of traffic and queueing at nodes lead to macroscopic dynamic phenomena [13–15]. In this context, the traffic of information packets in complex networks is a prototypal system for the theoretical study of traffic congestion phenomena arising with increased traffic load but limited network capacities to process it [13–16]. It has been understood that the occurrence of traffic jamming can be related to the network's structural properties, queuing rules, and the routing strategies that the nodes apply to transmit packets towards their predefined destinations. Therefore, to postpone the onset of jams, different methods have been investigated. Specifically, they aim at enhancing routing strategies [13,17–22] and changing the role of network's elements (nodes, edges, layers, communities) in the transmission process [3,19,23–27], as well as introducing different prioritising rules among packets [28,29].

Substantial research activity in the last two decades has been devoted to modelling the traffic of information packets with finite network capacities. It aims to develop a minimal model that can capture the mechanisms of traffic flow in different regimes: free flow, jamming transition and congested phases, respectively. Here, we restrict the discussion to our model originally developed and studied in [1,13,14,17]. In the model, each packet is monitored as it moves from the source node to its destination address as another node on the network, where it is delivered and removed from the process. When more than one packet appears at a node, the packets form a queue by order of arrival at that node. Each node has a finite queue length, and the packet at the top of the queue is processed first (LIFO queueing discipline). The node performs an in-depth search for the packet's destination address to transmit a packet. If the address is found in the searched depth, the packet is transmitted to the top of the queue of its neighbour along the shortest path towards the destination. Otherwise, it is sent to a random neighbour, who then repeats the search; see Section 2.2 for a more detailed description. In this context, the network structure is essential for how the searched depth covers the network's information horizon. For example, the efficiency of the next-neighbourhood search on the correlated scale-free graph leads to superdiffusion in the low-density traffic, as shown in [13]. The model implementation allows variants with different searched depths, the FIFO rule, and the constant density traffic, as shown in refs. [13,30] and [1], respectively.

By increasing the posting rate $R$, more packets are found in the traffic, leading to increased queue lengths and total traffic load. However, the average delivery balances the packet input until a critical rate $R_c$ is reached, at which this balance is lost and congestion occurs. It manifests in the steady increase in the network load and dramatic increase in the waiting times and travel times of packets, eventually resulting in packet loss (divergent travel time). As stated above, the critical rate $R_c$ and the nature of the transition to the congested phase strongly depend on the network structure and the efficiency of the search rule on that structure [1,13,14,30]. For example, with the above-described navigation rules with next-neighbour search, the congestion occurs via an *abrupt change of the order parameter*, defined as the time derivative of the network load averaged over different time windows [1]. Preceding the congested state, traffic load arises gradually, depending on the posting rate, navigation rules, and network structure. The maximum queue length in the networks with hubs is first reached on the hubs and is then gradually spread to their neighbouring nodes. The congested traffic is then characterised by a dramatic slowing down, where one packet leaves the hub's queue and only then can another one arrive at it. Therefore, exploring the nature of the fluctuations of a traffic load time series in pre-congested phases is of paramount importance and could be used as a warning sign of the approaching jamming transition. Such time series data are easily accessible in many real systems. Previous studies (see [1,13,14] and references therein) have shown how various statistical features, which are based on the individual packet's transport, change with the increased traffic load on different networks and search rules. Hence, the increasing traffic density manifests in the modified distributions of the travel time and waiting times of packets along their paths and changed distributions of the avalanches of active nodes. Another aspect of traffic jamming was captured by the geometrical representation of the traffic time series in [31], suggesting that the jamming is accompanied by complex transformations of the structure of the system's phase space.

Here, we focus on the nature of fluctuations in the network load time series for several posting rates preceding the traffic congestion, simulated on two different network structures. These representative network topologies are described below and in [1]; they appear to have efficient traffic flow for low (*Webgraph*) and high (*Statnet*) traffic density before the congestion occurs at the respective critical densities. Our analysis reveals certain persistent features of the traffic load fluctuations in the jamming regime on both network types, which are described by cyclic trends. Their multifractal fluctuations are described by the spectrum of generalised Hurst exponents with a similar broadening at the side of small fluctuations as the system approaches its respective jamming transition. At the same

time, such oscillations are accompanied by gradually-lost coherence in the network activity and the prevalence of the short-range correlations in the power spectra. These traffic-load time series features are early warning signs of approaching congestion, complementing the statistics of individual packets' queueing and travel times.

## 2. Substrate Networks and Traffic Model Rules

### 2.1. Properties of Two Prototypal Networks

We consider the traffic of information packets on two network structures shown in Figure 1. Their structural characteristics are given below and in [1], appearing to be well-optimised for efficient low-density traffic (*Webgraph*) and high-density traffic (*Statnet*), respectively. As described in [1], both networks are generated using the same sets of probabilities for the *preferential linking and preferential rewiring*, with the crucial difference that the *Webgraph* is a growing network, while *Statnet* is a fixed-size network with links added among pairs of nodes under the condition, preventing multiple links. The number of nodes $N = 1000$ and edges per node ratio is $E/N = 1$ in both networks.

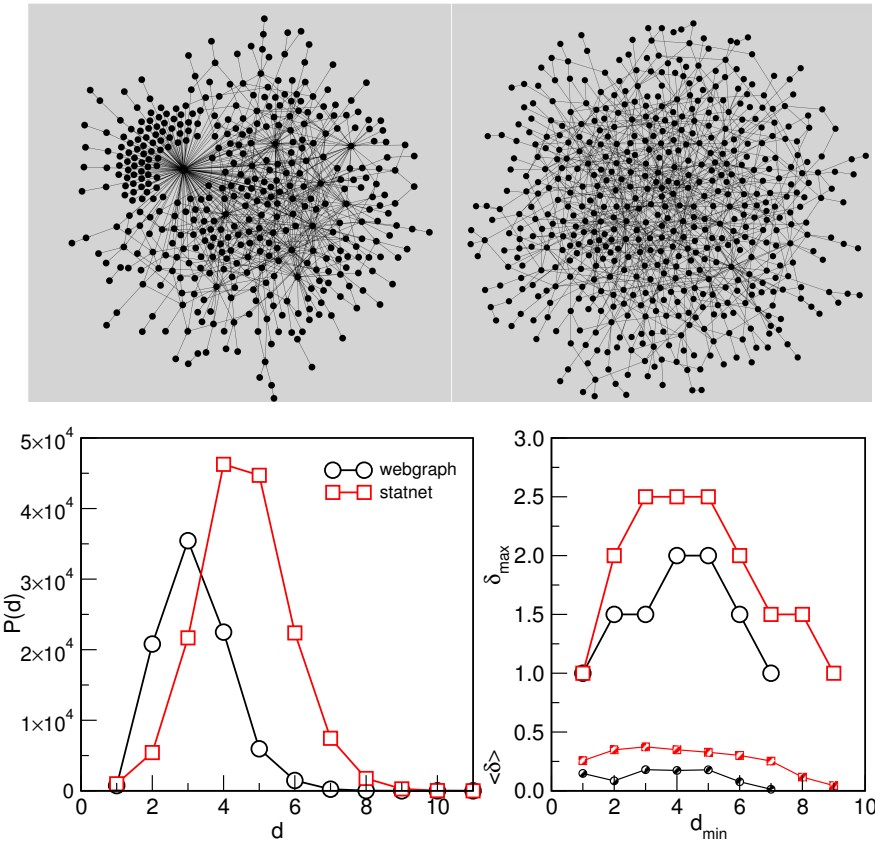

**Figure 1.** Networks' giant clusters on which traffic simulations are performed: *Webgraph* (**left**) and *Statnet* (**right**). See the text for these networks' properties. Lower panels: Distribution of the shortest-path distances $P(d)$ vs. distance $d$ between pairs of nodes, and hyperbolicity parameter $\delta_{max}$ vs. minimal distance $d_{min}$, see text, on these two graphs, as indicated in the legend.

The outcome is that the *Webgraph* is a *correlated scale-free graph*, the structure is statistically similar to the Web considered as a directed graph; see the original work in [32]. In addition, the *Webgraph* possesses disassortative degree correlations; see [1] for details. Meanwhile, the *Statnet* appears to have a much weaker organisation, a stretched-exponential profile of the node's degree ranking distribution, and the absence of any degree–degree correlations [1]. For completeness, some standard graph properties that might be relevant to the traffic on these networks are summarised below in Table 1. In addition, we determine the distributions of the shortest path distances between all node pairs in the related giant

clusters, as shown in the bottom panels in Figure 1, where we also demonstrate that these are hyperbolic graphs in the sense of the Gromov hyperbolicity criterion [33,34]. These figures show that the most probable distance between pairs of nodes on the *Webgraph* is $d = 3$, compared to $d = 4$ or 5 on the *Statnet*, and with a different hyperbolocity, as discussed below. Overall, despite a comparable average degree, the *Statnet* possesses a larger network diameter and a larger average shortest path, as well as a much smaller number of triangles and clustering coefficients than the *Webgraph*, cf. Table 1.

The network's hyperbolicity [35], or negative curvature in the graph's metric space (shortest paths), is highly relevant for different transport processes and more general diffusion phenomena [36,37]. In this context, the graphs with a small Gromov hyperbolicity parameter $\delta_{max}$ are particularly interesting [38]. Among them are some of the most efficient naturally evolved structures, from networks mapping the human brain structure [39,40], metabolic networks [41] and chemical graph structures [42,43], to online social graphs [44,45]. For a finite graph $G$, the hyperbolicity parameter $\delta_{max}$ is determined [36,39,46] by sampling a large number of 4-tuples of nodes $\{A, B, C, D\}$ and considering the ordered sums of their distances, for example, $\mathcal{S} \equiv d(A, B) + d(C, D) \leq \mathcal{M} \equiv d(A, C) + d(B, D) \leq \mathcal{L} \equiv d(A, D) + d(B, C)$. For a $\delta$-hyperbolic graph, there is $\delta(G)$ such that any four nodes of the graph satisfy the condition $\delta(A, B, C, D) \equiv \frac{\mathcal{L} - \mathcal{M}}{2} \leq \delta(G)$ From the triangle equality, we have that $(\mathcal{L} - \mathcal{M})/2$ is bounded from above by $d_{min}$, the minimal distance $d_{min} \equiv min\{d(A, B), d(C, D)\}$ in the smallest sum $\mathcal{S}$, which is then used to compute $\delta_{max}$ as the largest $\delta(G)$ found in all 4-tuples. The two networks considered in this work appear to belong to this category of graphs, cf. Figure 1 bottom right. The value of $\delta_{max} = 2.5$ in the *Statnet* suggests the occurrence of larger characteristic cycles [47] compared to $\delta_{max} = 2$ in the *Webgraph*.

**Table 1.** Network properties: the average degree $< k >$, clustering coefficient $< Cc >$, and path length $< \ell >$, the number of triangles, modularity, the network's diameter $D$ and the Gromov hyperbolicity parameter $\delta_{max}$.

| Network | $< k >$ | $< Cc >$ | No. Triang | $< \ell >$ | mod | $D$ | $\delta_{max}$ |
|---------|---------|----------|------------|------------|-----|-----|----------------|
| *Webgraph* | 3.439 | 0.175 | 192 | 3.196 | 0.497 | 9 | 2.0 |
| *Statnet* | 3.593 | 0.010 | 24 | 4.563 | 0.546 | 11 | 2.5 |

### 2.2. Traffic of Information Packets: Model Rules

As mentioned in the Introduction, the relevant parameters of the packet traffic model are the posting rate $R$, the maximum queue length of each node $H$, and the searched depth; here, we fix it with the next-neighbourhood (2-depth) search. In simulations, each packet is an object, the properties of which change in time; precisely, at each time step, the current location node of each packet is monitored, along with its position in the current queue and the waiting time before it leaves that node. The travel time of a packet is counted as the sum of waiting times at nodes along its path, and the corresponding distributions are determined; see the results in [1,14,30]. For this work, we sample time series, in particular, the number of posted packets $n_p(t)$, the number of active nodes moving a packet $n_a(t)$, and the total number of packets still in the traffic (the network load) $N_p(t)$. It is given by the sum of all queue lengths $Q_i(t)$ at time $t$, i.e., $N_p(t) = \sum_i Q_i(t)$, $i = 1, 2, \cdots, N$. Another relevant time series is the number of packets delivered $n_d(t)$ at time step $t$. Then the network's delivery rate is given as the time average $\langle n_d(t) \rangle$. For the above-described networks, we use $H = 1000$ and increase the posting rate $R$ to reach $R_c$ for the existing network; see below. The long time series up to $6 \times 10^5$ steps are produced, particularly for $R \lesssim R_c$, to ensure that the network is not congested.

The simulation of packet traffic starts with inserting the network structure as its adjacency matrix and setting the empty node's queue lengths $Q_i(t = 0) = 0$, at each node $i = 1, 2, \cdots, N$. Then, the information packets are created and navigated through the network according to the model rules [1,13,14,17,48], consisting of:

- *Posting*: At each time step *t*, each node can create a new packet with the probability *R*; another randomly selected node sets the packet's destination on the network's connected component; the created packet is added to the top of the node's queue;
- *Queueing*: If more than one packet is present at a node, they make a queue by order of arrival at that node, with a new arrival appearing at the top of the queue. The node's queue length at the time *t* is $Q_i(t) \in [0, H]$, where *H* represents the maximum possible queue length of each node;
- *Navigation*: Each node with a nonempty queue tries to move the top packet in its queue, i.e., we apply LIFO (last-in-first-out) queueing rule. The node performs a next-neighbourhood search for the destination address of the packet; if the address is found in the searched depth, the packet is delivered to the neighbour along the shortest path to the destination, else it is transferred to a random neighbour. If the neighbour queues are full, the packet waits for the next transmission opportunity;
- *Delivery*: Upon arrival at its destination, the packet is removed from the traffic.

An example of these time series on *Webgraph* is depicted in Figure 2 for the posting rate that exceeds the jamming transition. The figure shows that, despite the network's activity, the posting rate outbalances the delivery rate, resulting in the predominant increase of the total load $N_p(t)$. Consequently, these time series exhibit characteristic fluctuations, as shown in the right panel of Figure 2 with their standard deviation functions $F_2(n) \sim n^{H_2}$ plotted vs. the interval length *n*, where $H_2$ stands for the standard Hurst exponent. As explained above, the posting rate is a random process; meanwhile, we have $H_2 \sim 1$ for the increasing traffic load time series. At the same time, we see that the network's activity time series and the delivery rate change the slope towards a random process, $H_2 \sim 0.5$, suggesting a loss of the network's coherence when the jamming occurs. As was shown in [1], for these two networks and the 2-depth search rule that we also use here, the transition to the congested phase occurs via a jump (first-order phase transition) of the order parameter, defined as the time-averaged changes of the load $O \equiv \langle dN_p(t)/dt \rangle$. The critical values are $R_c \approx 0.4$ and $R_c \approx 0.8$ for the *Webgraph* and *Statnet*, respectively. In the following, we will analyse the power spectra and multifractal features of the traffic load time series for the posting rates *preceding* the respective critical rates. As stated above, the idea is to find some universal characteristics that can be used as signatures of the approaching abrupt transition to the congested phase.

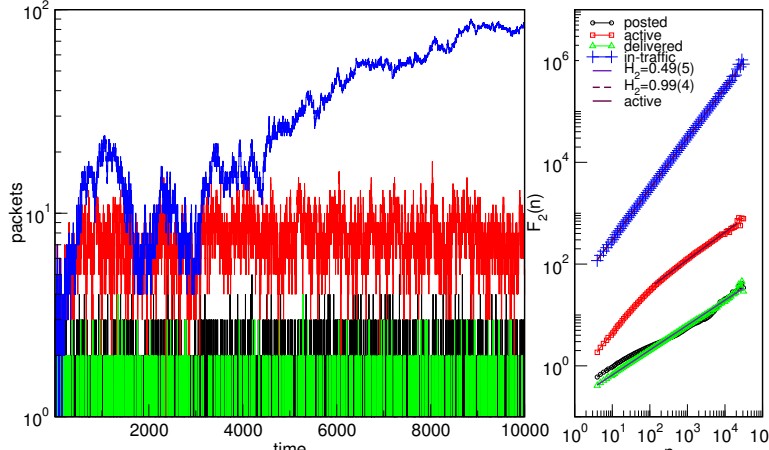

**Figure 2.** Example of time series of the number of created, active, still in traffic, and delivered packets monitored during the simulations of transport on *Webgraph* for the posting rate $R = 0.4 \gtrsim R_c$ (**left**). The standard deviations $F_2(n)$ fluctuation function of Equation (2) for $q = 2$, vs. time interval *n* for these time series (**right**); the same colour code applies to both panels.

### 3. Traffic Cycles and the Power Spectra of Load Time Series on *Webgraph* and *Statnet*

By increasing the posting rate in the range $R < R_c$ on a given network and using the same navigation rules, the average traffic load of packets gradually increases, with fluctuations compatible with the stationary time series, as shown in Figure 3 for the *Webgraph*, and Figure 4 for the *Statnet* structure. Due to a larger critical value $R_c$ in the *Statnet*, the corresponding network loads are higher than in the *Webgraph* before its congestion. On the other hand, the *Webgraph* equipped with the hubs and nodes correlation exhibits a cooperative transport for low posting rates, which is quantified with a power-law distribution of the avalanches of active sites [31]. On approaching the critical rate $R_c$, however, this cooperative functioning is gradually lost, and the size of active-site avalanches reduces to the one characteristic for a random process; see the detailed analysis in [31]. These changes of the network activity also manifest in the power spectra of the traffic load; cf. Figure 3 top left. In particular, the power spectral density exhibitis a decay with the frequency $f$ according to the power-law,

$$S(f) \sim f^{-\phi} , \tag{1}$$

where the exponent $\phi$ has a value slightly exceeding $\phi = 1$ for very low (but finite) posting rates. As shown in Figure 3, top left, the exponent increases and eventually reaches $\phi = 2$ when the steady increase of the load occurs. On the other hand, the power spectra of the loads on the network without hubs, cf. Figure 4, top left, possess two slopes even at low-density traffic; a short-range correlated part with $\phi = 2$ occurs in the high-frequency region and extends to the entire range of frequencies when the congestion occurs. Meanwhile, the low-frequency part exhibits an increasing exponent, eventually reaching $\phi = 2$.

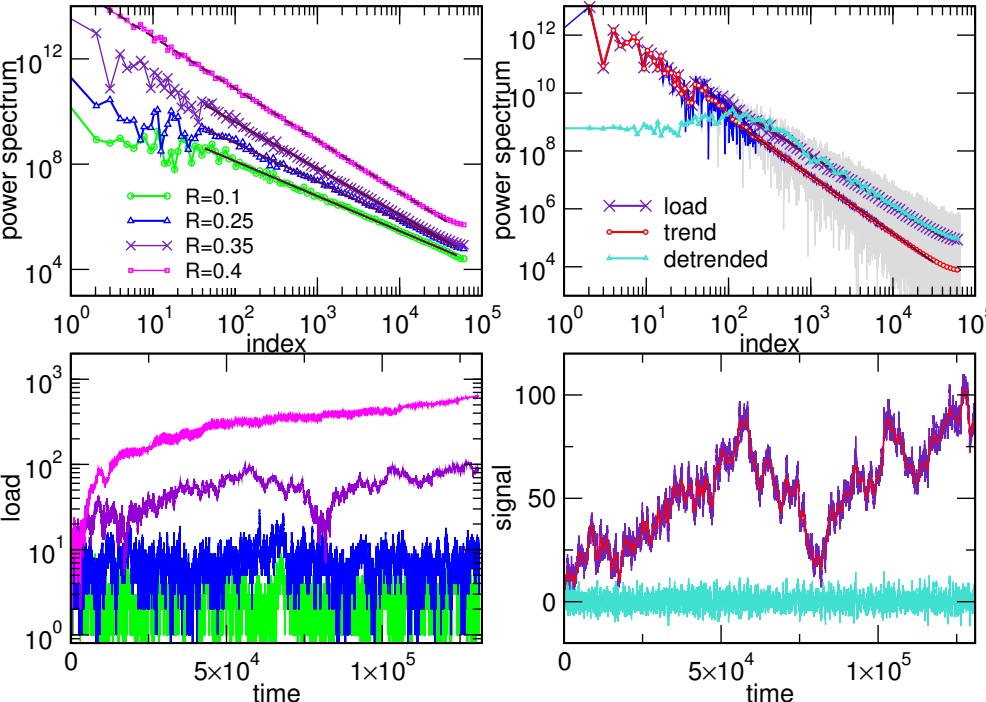

**Figure 3.** Traffic on the *Webgraph*: Time series of the network load (**bottom left**) and their power spectra (**top left**) for different values of the posting rate $R$ indicated in the top panel. The **bottom right** panel shows a close-up of the traffic load time series for the posting rate $R = 0.35$ with its cyclic trend (red line) and the detrended signal (cyan); the corresponding power spectra of these signals are shown in the **top right** panel, as indicated in the legend.

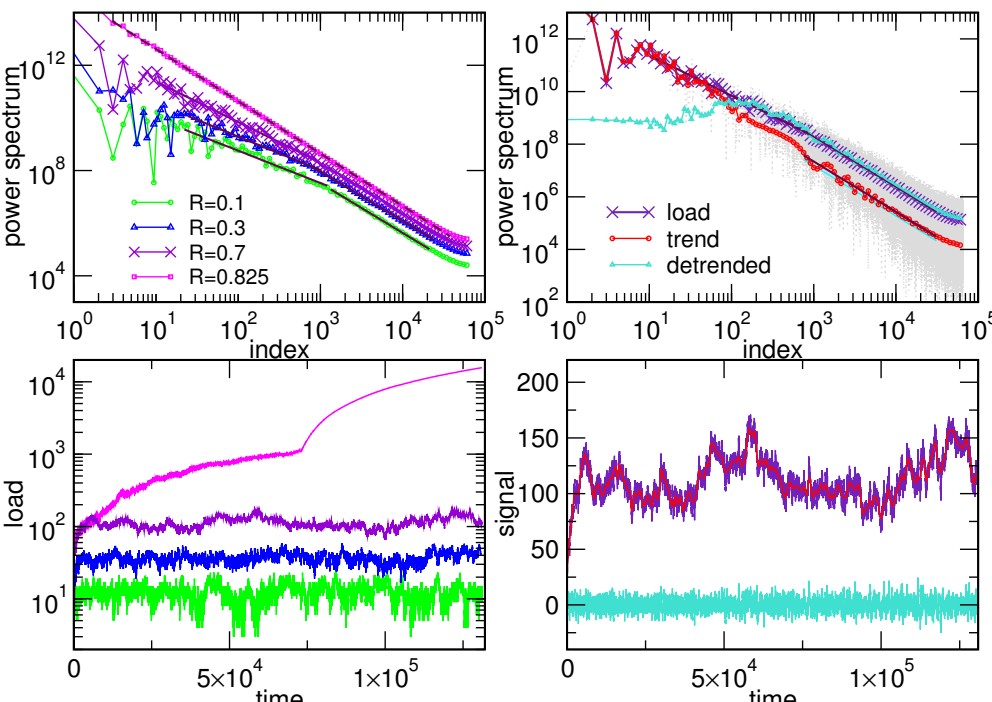

**Figure 4.** Same as Figure 3 but for the traffic load on the *Statnet*. The corresponding values of the posting rate $R$ are indicated in the top left panel. The traffic time series in the bottom right panel is for the posting rate $R = 0.7$, showing its cyclic trend (red line), the detrended signal (cyan), and the corresponding power spectra in the top right panel.

Another striking feature of these traffic load fluctuations preceding the jamming transition is the occurrence of an irregular *cyclical trend*, as is shown in Figure 3, bottom right, for the case of *Webgraph*, and in Figure 4, bottom right, for the traffic on *Statnet*. To determine these trends, we use the local adaptive detrending algorithm [49–51]. Specifically, based on the original work in [49] treating sunspot time series, the algorithm was adapted to treat different types of time series, from social network activity to the magnetisation fluctuations on the hysteresis loop [50,51]. The actual time series of the length $T_{max}$ is divided into segments of the length $2m + 1$, which overlap over $m + 1$ points. These intervals are enumerated as $k = 0, 1, 2, \cdots, k_{max} = T_{max}/m - 1$. The polynomial fits $y^{(k)}(mk + \ell)$ over $\ell = 0, 1, 2, \cdots; 2m$ points in each segment are determined. For $0 < k < k_{max}$, the trend $y_c(mk + i)$ over the overlapping points is determined by balancing the contribution of the polynomial in segment $k$ with that of segment $k + 1$ as $y_c(mk + i) = \frac{i}{m} y^{(k+1)}(mk + i) + \frac{m-i}{m} y^{(k)}(m(k + 1) + i)$, where $i = 0, 1, 2 \cdots, m$. Thus, the corresponding polynomial contribution in the overlapped region decreases linearly with the distance from the segment's centre. Meanwhile, the trend coincides with the actual polynomial fit in the initial $m + 1$ points in $k = 0$ and the final $m + 1$ points in $k = k_{max}$ segments. The linear interpolation suffices for the studied time series, and the parameter $m$ is adapted accordingly. As Figures 3 and 4 show, a complex structure of these cyclical trends appears to have many harmonics, depending on overall network load; see Section 4 for further analysis. The standard fluctuation function of the detrended signal saturates for time intervals $n > m$. As expected, the power spectrum of the fluctuations around the trend coincides with the signal's spectrum for the high-frequency region. Meanwhile, the trend shows the power spectrum with the exponent $\phi = 2$, cf. top right panels of Figures 3 and 4.

## 4. Mutifractality of the Traffic-Load Trends and Detrended Fluctuations

In the following, we analyse the traffic load trends for both low and high posting rates on both network structures. To apply the detrended multifractal analysis [52–54] of the signal's deviation from the local average, we construct the profile $Y(i) = \sum_{k=1}^{i}(C(k) - \langle C \rangle)$ of the time series and divide it into $N_s$ segments of length $n$. The process is repeated starting from the end of the time series $t = T_{max}$, resulting in $2N_s = 2Int(T_{max}/n)$ segments. The local trend $y_\mu(i)$ at each segment $\mu = 1, 2, \cdots, N_s$ is determined and the standard deviation around it, $F^2(\mu, n) = \frac{1}{n}\sum_{i=1}^{n}\left[Y((\mu-1)n+i) - y_\mu(i)\right]^2$, is determined. Similarly, $F^2(\mu, n) = \frac{1}{n}\sum_{i=1}^{n}[Y(N - (\mu - N_s)n + i) - y_\mu(i)]^2$ for $\mu = N_s + 1, \cdots, 2N_s$. Then, the fluctuation function $F_q(n)$ for the segment length $n$ and different values of the parameter $q \in [-4.5, 4.5]$ is determined as:

$$F_q(n) = \left(\frac{1}{2N_s}\sum_{\mu=1}^{2N_s}\left[F^2(\mu, n)\right]^{q/2}\right)^{1/q} \sim n^{H_q},\tag{2}$$

and is plotted against varied segment length $n \in [2, int(T_{max}/4)]$. The occurrence of the power-law sections on the lines for different $q$ are associated with the *generalised Hurst exponent $H_q$*, as indicated on the right-hand side of the expression (2). The case $q = 2$ corresponds to the standard deviation and the above-mentioned Hurst exponent.

In Figure 5, we show the fluctuation function $F_q(n)$ vs. segment length $n$ for the traffic trends detected in the load time series at low packet creation rate $R = 0.1$ on both network structures. Even though the creation rate is low, the traffic load is higher on the *Statnet* compared to the *Webgraph*. More importantly, the loads on both networks show a clear cyclical trend, as shown in the close-up in Figure 5. A detailed analysis of the fluctuation functions of these trends demonstrates their multifractal features on both networks. The corresponding generalised Hurst exponent $H_q$ is determined from the fitted region of these curves for different values of the parameter $q$. The results are summarised in the right panels of Figure 6. Notably, the large-scale fluctuations, which are captured by the fluctuation function for $q > 0$, are almost mono-fractal and compatible with the Hurst exponents $H_q \sim H_2 \sim 1$. Theoretically, this value of the standard Hurst exponent is observed in the class of stable Levy processes with limited range, in which small and large jumps are present. In contrast, the small-scale fluctuations corresponding to $q < 0$ need to be amplified with a whole spectrum of the exponents in the range $H_q \in [1.25, 2.5]$. A similar feature persists for the larger values of posting rate, including the one just before the congestion, i.e., $R = 0.35$ on *Webgraph*, and $R = 0.7$ on *Statnet*. The corresponding fluctuation functions are shown in the left panels in Figure 6, exhibiting multifractal properties in a wide range of time scales $n$. In addition, in both networks, a further broadening of the spectrum at the $q < 0$ side occurs with $H_q \lesssim 2.8$ as the system approaches the respective jamming transition. Hence, the increased number of higher harmonics in these cyclical trends appears as a robust feature of traffic jamming on different network structures; see also the Section 5.

The fluctuations around these trends are also investigated, as shown in Figure 7 for both networks and the two representative posting rates. Notably, these fluctuations are only weakly multifractal, specifically in the range $q < 0$, where the width $\Delta H_{q<0} \lesssim 0.4$, virtually independent of the posting rate $R$. Whereas, in all cases we find $H_{q>0} \sim H_2 \sim 1$, suggesting that these fluctuations still maintain temporal correlations characteristic of the whole signal.

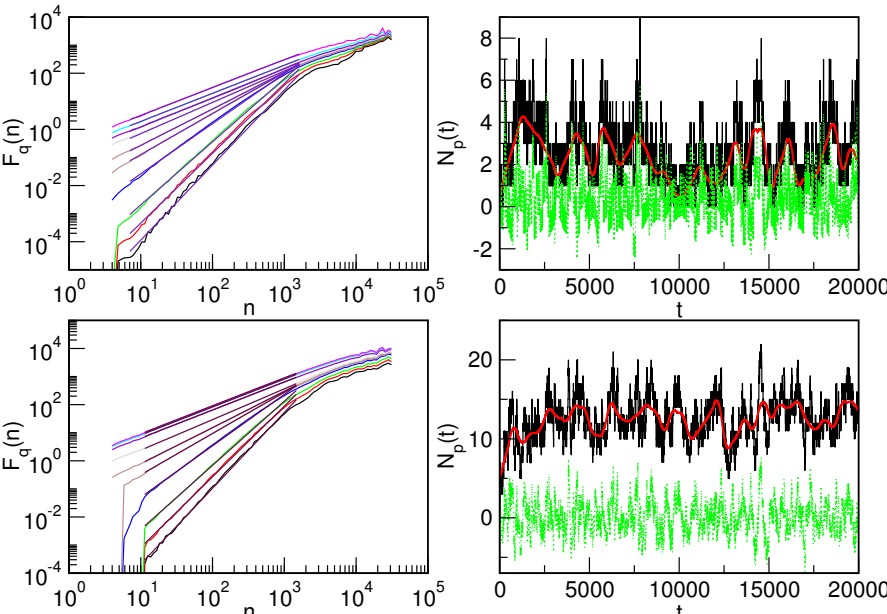

**Figure 5.** For the fixed packet generation rate $R = 0.1$ in the *Webgraph* (**top raw**) and *Statnet* (**bottom row**): left panel shows the fluctuation function $F_q(n)$ vs. $n$ for $q \in [-4.5, 4.5]$, of the traffic load's trend, which is shown by the red line in the corresponding right panel. The segment of the load time series $N_p(t)$ vs. $t$ is demonstrated by the black line, while fluctuations around the trend are shown as the green line. The straight lines indicate the fitted segments of the fluctuation function curves corresponding to the generalised Hurst exponent.

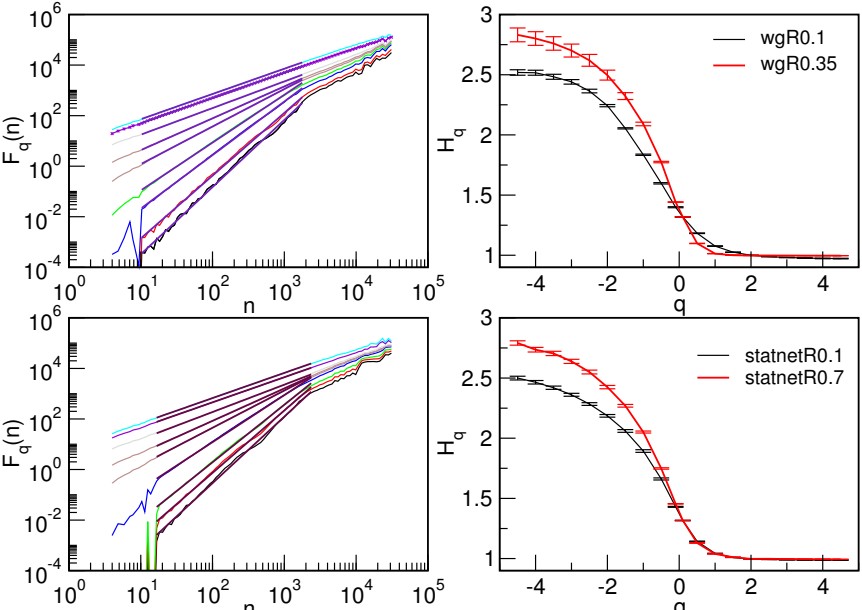

**Figure 6.** Left panels: The fluctuation function $F_q(n)$ vs. $n$ for the traffic load trends with jamming in pre-congestion flow, at $R = 0.35$ in *Webgraph* (**top**) and $R = 0.7$ in *Statnet* (**bottom** panel). The generalised Hurst exponent $H_q$ vs. amplification parameter $q$ for two posting rates, indicated in the legends for *Webgraph* (**top**) and *Statnet* (**bottom** panel).

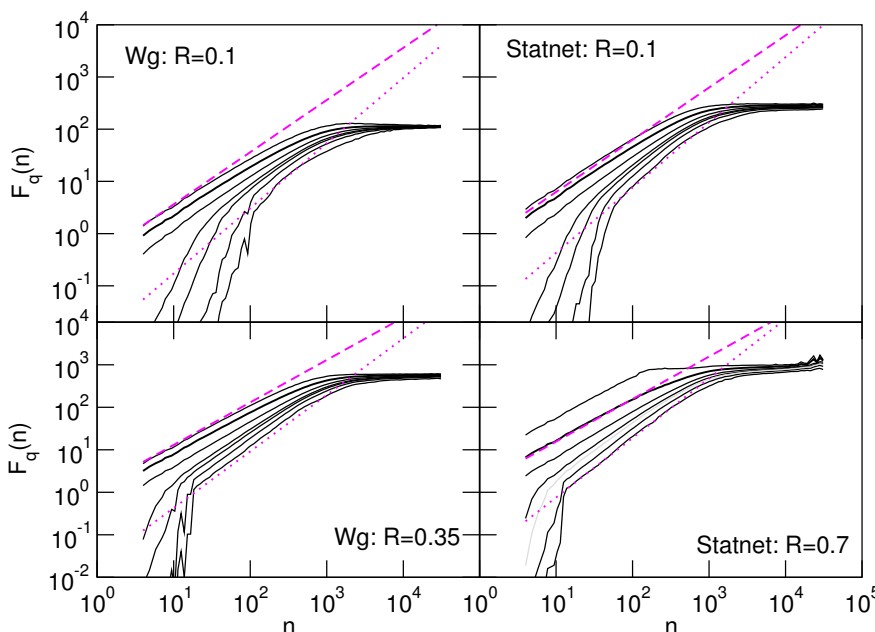

**Figure 7.** $F_q(n)$ vs. $n$ for the fluctuation around identified trends studied in Figures 5 and 6, for different networks and posting rates, as indicated in the legend. The slopes of the dashed and dotted lines are $1.02 \pm 0.05$ and $1.33 \pm 0.07$, respectively. Different lines in each panel correspond to $q = 4, 2, 0.5, -0.5, -1, -2, -4$, top to bottom.

## 5. Discussion and Conclusions

Using the previously developed model of the information packets' transport with search and queuing [1,13,14,17] on structured networks, we have investigated the nature of the traffic load fluctuations for densities below the transition into the congested phase. We have considered the data simulated with the model rule with a 2-depth search strategy and the fixed maximum queue lengths at all nodes [1,13,14]. Apart from the limited full queue lengths at all nodes, the network structure influences the packet traffic in various ways. It manifests in the efficiency of the packets' delivery, the distribution of their travel times and waiting times in queues, and the jamming densities, primarily depending on how the routing strategies adhere to the network structure, as previous studies showed [30]. In this context, the two networks considered in this work, *Webgraph* and *Statnet*, are good representatives of the structures that are 'optimal' for low and high traffic density, respectively [1]. They experience jamming via a 1st order phase transition at traffic densities related to different posting rates $R_c$, as stated above; see Figure 13 in reference [1].

Our main findings are that, despite considerable differences in the average packet densities resulting from the network's structural characteristics, traffic loads' global fluctuations close to jamming exhibit certain universal features. They are best captured by prominent cyclic trends in the traffic loads and their multifractal features, depicted in Figures 5 and 6. On the other hand, the fluctuations around the respective trends are only weakly multifractal, as shown in Figure 7. Considering different posting rates, i.e., traffic densities on the respective networks, such cycles may occur as soon as the density is high enough to cause packet queueing at different nodes. The network attempts to clear queues in a series of actions that spread to the neighbouring nodes. Such coordinated activity of nodes reduces the total load, after which it starts building up again with queueing at the busiest nodes. By approaching the jamming transition, the network's efficiency for packet delivery is reduced. This situation results in gradually reduced coherence and an increase of small variations of the load. Accordingly, the load's power spectra exhibit long-range temporal correlations for a wide range of low frequencies; cf. Figures 3 and 4. Meanwhile, the short-range correlations dominate when the congestion starts. A detailed analysis of the

trend's fluctuations revealed its multifractality, in particular for the small-scale fluctuations captured with the spectrum of the generalised Hurst exponents $H_{q<0}$, as shown in Figure 6 in both networks. As Figure 6 shows, the diversity of small fluctuations increases on approaching the jamming transition, which leads to a broadening of the $H_{q<0}$ spectrum.

It should be noted that the studied cyclic trends in traffic loads on networks are related to the system's states (phase space). Thus, they differ from the more familiar spatio-temporal evolution waves of localised groups of vehicles that appear as solutions of differential equations in macroscopic models of high-density traffic [4]. These robust characteristics of traffic load fluctuations can be used as warning signs for traffic congestion. They can be detected directly from the time series of network loads and complement the prominent statistical signatures, which require more detailed information about individual packets and the activities of local nodes in a heterogeneous network. Similar properties can be expected for the traffic loads of many other transport systems, provided that the mapping to the network geometry [8] properly takes into account the key elements of the underlying stochastic processes.

**Funding:** This research was funded by The Slovenian Research Agency under the funding number program P1-0044.

**Data Availability Statement:** Not applicable; this is a theoretical study, all data are model simulated.

**Conflicts of Interest:** The author declares no conflict of interest.

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
