# Peer review of "Cyclical Trends of Network Load Fluctuations in Traffic Jamming"

_2673-8716, doi:10.3390/dynamics2040026_

Round 1

Reviewer 1 Report

The manuscript presents the nature of traffic load fluctuations using the existing model of information packets transport on structural networks. The work is interesting. However, the presentation is somewhat not well-organized. This makes the manuscript difficult to read. Some suggestions for improvement are given as follows. 

It is better to remove mathematical notations (lines 59-76) from the Introduction, and add more verbal descriptions to explain the features and goal of the manuscript distinct from other works.  Since many mathematical notations are given in the manuscript, the author may list all those notations as a nomenclature.  

Please explain the meaning of the notation P(d) in Fig. 1, and the physical meaning of F_2(n)~n^H_2 in Fig.2.  Please describe the physical implications of Figs. 1-6 in detail.  The derivation of eqn. (2) needs to be given.  

Author Response

We thank the Reviewer for his time and valuable suggestions to improve the presentation of results.

In the revised manuscript, we have rephrased parts of the Introduction to describe the traffic jamming and the focus of the present work, pointing out the distinction from the previous results. We moved the definitions of the related quantities to section 2, subsection regarding the traffic model rules.

In Fig1, the distribution P(d) refers to the shortest-path distances d between all pairs of nodes on the considered graph. We expanded the figure caption to stress “between all pairs of nodes”. Note that we show histograms (not a normalized distribution) for both considered networks to demonstrate their structural differences, as shown in Table 1 and commented in the text.

F_2(n) is the standard deviation function for the time interval length n, which is known to define the Hurst exponent. Given the generalized eq.(2), it corresponds to the case q=2, as we now state in the caption of Fig.2

Physics insights of different Figures 1-6 are commented on in the text. Please see the expanded text referring to Fig1 on p3;

Regarding Fig2, we emphasizethat the text on p5 is commenting in detail on the results shown; please see lines 176 till the end of section 2 and the figure caption.

Fig3 and Fig4, referring to Webgraph and Statnet, respectively, contain network load time series for different posting rates R in the range from small loads (free flow) to the respective jamming rate Rc. Power spectral densities of these signals are shown in the separate top panels, exhibiting the differences in the long-range correlations discussed in the text; see eq.(1) and the related text on p6. A close-up segment of the time series close to jamming is shown in the right panel, exhibiting a cyclic trend with a lot of small harmonics, and detrended signal. Properties of these signals are then analysed in section 4.

We have expanded the part of the text referring to the determination of the cyclic trend by the adaptive detrending algorithm, which is now described in more detail. Please see the highlighted text preceding section 4.

Figs 5 and 6 show the fluctuation functions F_q(n) of the above-determined signals for low and high posting rates in two network types. The resulting generalized Hurst exponents H_q are summarized in Fig6, right panels. They show that in both networks, the traffic load fluctuations at jamming are dominated by the occurrence of cyclical trends that have a complex structure, including a lot of harmonics; see also added comment on p8.

Reviewer 2 Report

-more information is needed for : local adaptive detrending algorithm useful for trends determination (p6,line: 194)

-a short description of the computing environment infrastructure for the experiments would be helpful

Author Response

We thank the Reviewer for his time and valuable suggestions, which help us revise the text of the paper accordingly.

More precisely, prompted by the Reviewer, we have added a detailed description of the adaptive detrending algorithm, which is motivated by the original work in Ref.[50]. Please see the added text, highlighted on p6-7.

As regards computing resources, the related simulations are for the network’s size N=1000 nodes and the maximum queue length per node H=1000 packets, which do not require large memory. For this purpose, we sample long time series (up to 600.000 time-steps) to ensure the congestion is absent. The simulations are done on the high-performance computing facilities at the Theoretical Physics Department, Jozef Stefan Institute.

Round 2

Reviewer 1 Report

No other comments.